# Laser-Assisted Selective Fabrication of Copper Traces on Polymers by Electroplating

**DOI:** 10.3390/polym14040781

**Published:** 2022-02-17

**Authors:** Vitalij Fiodorov, Karolis Ratautas, Zenius Mockus, Romualdas Trusovas, Lina Mikoliūnaitė, Gediminas Račiukaitis

**Affiliations:** 1Department of Laser Technologies, Center for Physical Sciences and Technology, Savanoriu Ave. 231, LT-02300 Vilnius, Lithuania; karolis.ratautas@ftmc.lt (K.R.); romualdas.trusovas@ftmc.lt (R.T.); g.raciukaitis@ftmc.lt (G.R.); 2Department of Chemical Engineering and Technology, Center for Physical Sciences and Technology, Sauletekio Ave. 3, LT-10257 Vilnius, Lithuania; zenius.mockus@ftmc.lt; 3Department of Organic Chemistry, Center for Physical Sciences and Technology, Sauletekio Ave. 3, LT-10257 Vilnius, Lithuania; lina.mikoliunaite@ftmc.lt

**Keywords:** electroplating, laser, graphene, polyimide, copper, selective, MID, traces, polymer, resistance

## Abstract

The selective deposition of metals on dielectric materials is widely used in the electronic industry, making electro-conductive connections between circuit elements. We report a new low-cost laser-assisted method for the selective deposition of copper tracks on polymer surfaces by electroplating. The technique uses a laser for the selective modification of the polymer surface. The electrical conductivity of some polymers could be increased due to laser irradiation. Polyimide samples were treated using nanosecond and picosecond lasers working at a 1064 nm wavelength. An electro-conductive graphene-like layer was formed on the polymer surface after the laser treatment with selected parameters, and the copper layer thickness of 5–20 µm was deposited on the modified surface by electroplating. The selective laser-assisted electroplating technology allows the fabrication of copper tracks on complex shape dielectric materials. The technology could be used in the manufacturing of molded interconnect devices (MID).

## 1. Introduction

Recent developments in semiconductor technologies led to the miniaturization of various electronic devices. Alongside this growing trend, the vast rise of flexible electronics takes place. New methods for fabricating conductive tracks on flexible materials are constantly being developed to implement flexible and 3D electronics.

The technologies of the selective fabrication of copper tracks on dielectrics show massive potential for current and future electronics applications. There are several laser-assisted techniques for producing copper tracks on dielectric materials. One of the most common processes is laser direct structuring (LDS). LDS is a two-step method, where initially, an electric pattern is written by laser on the surface of the dielectric material. Furthermore, the material is immersed into the electroless copper bath, where copper deposits the laser-activated patterned area [1]. The disadvantage of this method is that special materials must be used, doped with inorganic metallic compounds for laser activation. This leads to an increase in the process price; also, the components with metallic impurities are not suitable for some applications. Another method is laser-induced forward transfer (LIFT). The principle of this technique is that thin donor material is placed above the specimen, and the laser exposition transfers the material from the thin donor to the sample [2]. The disadvantage of this technique is weak adhesion between the transferred donor and the substrate. Moreover, very little research has been conducted with 3D substrates. Laser-induced surface activation (LISA) is a three-step process: firstly, the specimen is laser-treated in distilled water; furthermore, it is immersed in the palladium solution to activate the treated area; and finally, the sample is copper-plated using an electroless plating method [3]. A novel three-step method is the selective surface activation induced by a laser (SSAIL) [4]. Initially, a pattern is written by a focused laser beam on the specimen surface. The further step is the chemical activation of laser-modified areas. Finally, metal is deposited on the laser-patterned area by electroless plating. The method offers high plating selectiveness and good copper adhesion. Furthermore, there is no need for special additives. Therefore, copper tracks could be produced on plastics and glass, and other dielectric materials. However, the electroless plating process is slow. Moreover, the instability of the plating bath strongly affects deposition quality. Currently, only the LDS method is used in manufacturing electronic devices.

Galvanic or electroplating is used to metalize the whole surface of a specimen made of polymer. The process requires the electro-conductive surface of the material [5]. However, the use of electro-conductive polymers and conductive composites prevents the spatial selectivity of the plating. Electroplating polymers are feasible, in some cases, by adding an initial electroless plating layer of nickel or copper to provide a conductive surface on the plastic part [6]. Laser irradiation could initiate a transformation of isolating polymer to conductive carbonaceous material.

During the past decade, graphene gathered extensive attention from research communities and the electronic industry. This is because this 2D material possesses a set of peculiar physical properties [7], making it a strong candidate for broad applications in electronics [8,9,10,11]. Moreover, from the various graphene production methods being developed, laser-based ones can offer production flexibility and scaling options. Laser-induced graphene (LIG) has already been proven as a suitable material for sensors and energy storage [12,13,14,15,16,17]. There are a few approaches for forming LIG. One of them is based on graphene oxide (GO) reduction with laser irradiation [14,18,19,20,21,22,23,24]. This method involves a GO precursor, which is prepared using chemical methods. On the other hand, producing graphene structures from commercial polymers [25,26] is more attractive, as no additional steps are needed to form GO coatings.

It is known that LIG can be formed in various organic materials [27,28,29] and polymers. For example, various groups reported the successful formation of LIG in polyimide using both far-infrared CO_2_ [13,16,30,31,32,33,34] and ultrashort pulsed lasers working from IR to UV wavelengths [35,36,37].

This research presents an alternative cooper tracks formation method, including LIG formation in the commercial polymer, for the first time. Our laser-assisted technology consists of two main steps: firstly, the electric circuit is laser-written on a dielectric surface. Then, the sample is immersed into the electrolytic solution, where laser-treated areas are copper-deposited (Figure 1). 

Surface conductivity measurements and Raman spectroscopy were applied to characterize the laser-modified polymer surface. Then, electroplating parameters were varied to obtain a uniform copper layer on laser-treated areas. The main advantage of the method is that direct electroplating on polymers allows an easy, selective, and low-cost metal deposition process. Direct electroplating is a much faster copper deposition method compared with other alternative technologies, based on electroless plating, like LDS [1] and SSAIL [4]. Additionally, the method could be used for the production of copper traces on 3D complex shape materials, where conventional methods, such as photolithography, are not suitable. The disadvantage of this method is that it could be used on certain groups of polymers, of which the surface conductivity could be increased after laser treatment. 

## 2. Materials and Methods

### 2.1. Materials

Polyimide Kapton^®^ films from DuPont with a thickness of 127 μm were used in our experiments. All the samples were cleaned with ethanol (Sigma-Aldrich, Burlington, MA, USA) before the experiments.

### 2.2. Laser Treatment

Two different lasers were used for sample treatment in our experiments. First, a nanosecond solid-state laser Baltic HP (Ekspla, Lithuania) was operating at the fundamental wavelength of 1064 nm. The pulse duration of this laser was 10 ns, and the pulse repetition rate was tuned from 10 to 100 kHz. The maximum pulse energy of 115 μJ was used. Another was a picosecond solid-state Atlantic (Ekspla), operating at 1064 wavelength. This laser generated 10 ps pulses at a repetition rate from 100 to 400 kHz. Pulse energies of up to 85 μJ were used.

For laser beam positioning, a SCANgine (ScanLab, Puchheim, Germany) galvanometer scanner was used. Beam positioning speed was 100 mm/s during our experiment. Telecentric F-theta objectives with 160 mm and 100 mm focal lengths were used to control the laser spot size. Samples were processed at the focal plane and with a defocused laser beam when a sample was placed up to 12 mm below the focal plane. Enlargement in beam diameter on the sample surface caused a lower energy density; however, the exposure time was longer due to the enlargement of the beam diameter and a fixed translation speed. The number of pulses per spot could be varied through defocusing. For example, when fabricating at the focal plane, where beam diameter was 52 μm, 52 pulses hit the same spot. On the other hand, fabricating 8 mm below the focal plane, as many as 480 pulses hit the same spot at a constant speed and repetition rate, due to beam enlargement. The irradiation dose was evaluated by multiplying laser fluence by the number of pulses per beam spot area. 

The optimal processing parameters, like scanning speed or repetition rate, were found during the testing fabrications, and were kept constant in the later experiments. The matrixes of laser-treated areas were used in a search for optimal parameters. The principle of this method is that the matrix of small squares (5 × 5 mm^2^) was fabricated on the sample by changing only the scanning speed value when processing squares in a row and changing the laser pulse energy value when fabricating the next square in a column. Multiple matrixes of squares were fabricated using different pulse repetition rates. Furthermore, sheet resistances of the laser-treated surface areas were measured. A combination of optimal processing parameters was selected according to treatments, after which surface sheet resistances tended to decrease. 

### 2.3. Electroplating

The metal deposition was performed using galvanic plating. Laser-treated Kapton PI film was connected to the cathode terminal for copper deposition, and a copper rod (Cu) was connected to the anode terminal. Electroplating was performed in the electrolytic solution, which consisted of sulfuric acid (H_2_SO_4_) 50 g/L, copper sulphate (CuSO_4_) 200 g/L, and deionized water (H_2_O). Plating speed at a current density of 10 mA/cm^2^ was 0.22 µm/min (Figure 2).

### 2.4. Sheet Resistance Measurement

Sheet resistance measurements of laser-treated areas were performed using the four-probe method. The principle of this method is that four probes were aligned in a row, with a distance of 1 mm between probes. While the current passed through external probes, the inner probes were used to measure the voltage. These values allowed us to calculate the surface resistance. Sheet resistance measurements were performed using the source meter (Keithley 2602A, Keithley, Cleveland, OH, USA) with the measurement software (TSP^®^ Express, Keithley, Cleveland, OH, USA) (Figure 3).

### 2.5. Surface Morphology Analysis

Microscopic observation of fabricated sample surface was done using an optical microscope (Olympus BX52, Olympus, Tokyo, Japan) with a CCD camera. Objectives with magnification from 5× to 50× in bright and dark field modes were used. In addition, surface roughness measurements were performed using a S Neox optical profilometer (Sensofar, Terrassa, Spain).

## 3. Results

### 3.1. Laser Fabrication at the Focal Plane. Picosecond vs. Nanosecond Regime

In the initial part of the experiment, Kapton PI films were placed at the focal plane of the laser beam and treated with the nanosecond and picosecond lasers at 1064 nm. Rectangles with edge lengths of 10 mm and 20 mm were fabricated in the scanning direction along the minor edge. The treated area was obtained by fabricating with partly overlapping laser tracks (hatch = 30 µm and laser spot diameter at the focal plane—52 µm). The pulse repetition rate was 100 kHz, and the scanning speed was 100 mm/s. Irradiation doses were increased by increasing the laser power. All other parameters were kept constant. 

Changes in the sheet resistance appeared when the irradiation doses were higher than 20.41 J/cm^2^ in the ns-regime and 17.75 J/cm^2^ in the ps-regime. The sheet resistance of the laser-treated surface started to decrease at those values, and soot began to appear on the surface of samples (Figure 4). 

The lowest achieved sheet resistance was 354 Ω/square during the nanosecond laser irradiation with 184 J/cm^2^, and one order lower values were obtained after the picosecond laser irradiation. At this regime, the lowest sheet resistance was 30 Ω/square when the irradiation dose was 130 J/cm^2^. 

Irradiation with higher doses led to the damage of the sample. The treated surface became cracked and improper for further use. Treatment repeatability was better using the ps-laser treatment. Thin PI samples were deformed using the ns-regime due to excessive heating at irradiation doses higher than 118 J/cm^2^. Therefore, the experiments were later continued only using the ps-regime.

### 3.2. Fabrication at the Non-Focal Plane with 1064 nm

The second processing method of Kapton PI films was performed with the picosecond laser when the sample was placed out of the focal plane and treated with a defocused laser beam. Different beam spot diameters on the sample surface were adjusted by changing the sample position relative to the laser beam focal plane. The specimen was moved below the focal plane as much as 12 mm, and that caused the spot size to change from 52 μm at the focal plane to 754 μm at 12 mm below the focal plane (Figure 5). All other process parameters, including laser power, repetition rate or scanning speed, were kept constant during this experiment. Only the sample position relative to the focal plane was changed.

A set of experiments was performed with a 1064 nm wavelength. The sample was placed at the focal plane initially. Later, the sample was moved down below the focal plane by 3 mm. The irradiation dose in this distance was from 130 J/cm^2^ (at the focal plane) to 17.75 J/cm^2^ (3 mm below the focal plane). Soot was formed on the sample surface in this fabrication range (Figure 6a). Strong dependence of sheet resistance and applied irradiation dose were observed after fabrication in this defocus range. A thicker layer of soot formed when treated with a high irradiation dose and, accordingly, surface sheet resistance was lower. Sheet resistance was as low as 30 Ω/square when treated at the focal plane with 130 J/cm^2^ irradiation dose, and 20 MΩ/square when treated at 3 mm below the focal plane with a 17.75 J/cm^2^ irradiation dose. 

The fabricated area started to appear grayish while moving the sample down beyond 3 mm. In the range from 3.25 mm to 4.75 mm below the focal plane, laser irradiation dose in this distance was from 15.5 J/cm^2^ (3.25 mm below the focal plane) to 11.4 J/cm^2^ (4.75 mm below the focal plane). After fabrication with these irradiation doses, part of the fabricated area was covered by gray structures (Figure 6b). Those structures were with a sheet resistance of 70–80 Ω/square. However, adhesion between the formed cluster and sample surface was weak. As a result, part of the structures dropped off when the sample was bent over a few times. 

Further moving the sample out of focus led to the formation of smooth, gray graphite-like structures, as we suggest laser-induced graphene (LIG) (Figure 6c). Those structures were formed when fabricating in the distance range from 5 mm to 12 mm below the focal plane, and the irradiation dose in this range varied from 10.76 J/cm^2^ (5 mm below the focal plane) to 4.42 J/cm^2^ (12 mm below the focal plane). Furthermore, the sheet resistance was constant on the whole treated area in this defocus range, and it was 8 Ω/square. Thus, the achieved sheet resistance value was the lowest in our experiments.

The PI surface became unaffected after the laser irradiation when the sample was placed 12 mm below the focal position. This was because energy density was too low to affect the sample. The fabrications were performed when the samples were lifted above the focal plane up to 12 mm in height as well. The results were the same.

Three different modifications of PI surface after irradiation with the laser could be distinguished after treatment at various distances from the focal plane with 1064 nm (Figure 7). Soot was formed on the surface of the sample when fabricating in the range around the focal plane (0 mm to 3 mm below the focal plane, the irradiation dose was from 130 J/cm^2^ to 17.75 J/cm^2^). Clusters of gray color structures formed when the polyimide sample was placed from 3.25 mm to 4.75 mm below the focal plane (irradiation dose 15.5 J/cm^2^–11.4 J/cm^2^). Finally, LIG occurred when the sample was placed from 5 mm to 12 mm below the focal plane (irradiation dose 10.76 J/cm^2^–4.42 J/cm^2^).

We could not obtain LIG structures on the PI surface when irradiating at the focal plane. Nevertheless, the irradiation dose was lowered to the same values by simply decreasing the laser power. The occurrence of LIG structures on the PI sample strongly depends on lasing time and how many laser pulses will affect the same spot on the sample.

### 3.3. Raman Spectra Investigation

Raman microscope Alpha300R from WITec, Germany was used to investigate alterations in polyimide film after irradiation with focused and defocused beams of the ps-laser. The 532 nm excitation wavelength and 100 s integration time were applied during the measurements. Defocusing the laser beam led to changes in irradiation dose applied to the PI sample surface. The dose varied from 130 J/cm^2^ at the focal plane to 4.42 J/cm^2^ at a distance of 12 mm below the focal plane at 1064 nm. 

The Raman spectra of Kapton PI films, treated with the ps-laser beam, exhibited sharp, high-intensity 2D peaks, evidencing graphene formation during laser treatment. Various Raman spectra parameters indicated the formation of high-quality LIG in a certain range of doses. A comparison of FWHM values for the main spectral bands D, G, and 2D is presented in Figure 8a. These bands exhibit a narrowing, starting from the irradiation dose of 5.4 J/cm^2^, reaching a sharp FWHM minimum at 7.34 J/cm^2^. At that irradiation dose, FWHM(D) = 38.3 cm^−1^, FWHM(G) = 24.4 cm^−1^ and FWHM(D) = 48.7 cm^−1^. Above that particular dose, all spectral bands broaden significantly slower, increasing the irradiation dose. A decrease of FWHM of the D and G corresponds to decreasing disorder in material [38]. FWHM of the 2D band also tends to decrease when the number of graphene layers decreases [39,40]. Raman spectrum obtained at the optimal irradiation dose of 7.34 J/cm^2^ is shown in Figure 8b. All spectral bands prominent to graphene are present: band D with peak center position at 1350 cm^−1^, band G at 1579 cm^−1^ and 2D at 2691 cm^−1^. It is worth noting that the D band, representing the structural defects, possesses particularly low intensity compared to the G band. Intensity ratios I(2D)/I(G) and I(D)/I(G) are recognized markers for the investigation of graphene.^19^ In this case, the I(2D)/I(G) = 0.4 and I(D)/I(G) = 0.1. A more detailed dependence of these ratios is shown in Figure 9.

Structural defects of graphene can be assessed using the information provided by Raman spectra. Line defects can be identified by the average crystallite size $L_{\alpha }$ [41]. The spatial confinement of the crystallites determines the width of the G band, and $L_{\alpha }$ can be approximately determined by the equation [42]:(1)$$L_{\alpha }=\frac{l_{C}}{2}\ln\left[\frac{C}{FWHM\left(G\right)-FWHM\left(G_{0}\right)}\right],$$
where the coherence length *l_C_* = 32 nm, *C* = 95 cm^−1^ and *FWHM*(*G*_0_) is the width of the *G* band of undoped pristine graphene (15 cm^−1^).

The LIG crystallite size was evaluated at various irradiation doses using this equation. The dependence is plotted in Figure 9. The apparent maximum crystallite size was 37 nm when a 7.34 J/cm^2^ irradiation dose was applied. This dose value coincides with the narrowest FWHMs for spectral bands D, G and 2D (Figure 8a). At this dose, the minimum of I(D)/I(G) = 0.1. On the other hand, the maximum value of I(2D)/I(G) = 0.58 ratio was reached at a lower irradiation dose of 6.72 J/cm^2^.

To conclude, spectral band intensity ratios, together with the estimated crystallite size and FWHM of prominent spectral bands, indicate that the structural properties of LIG can be tuned by varying laser irradiation doses. LIG formation with minimal structural defects and fewer layers of graphene structures with the largest crystallites is in the range of irradiation doses 5.73–13.4 J/cm^2^. Furthermore, the formation of the best quality LIG—closest to spectral properties of graphene—is at the irradiation dose of 7.34 J/cm^2^.

### 3.4. Modeling of the Heat Conduction after Laser Pulses Exposition to a Substrate

The simulation of temperature dynamics after laser irradiation was carried out using COMSOL Multiphysics software. Transient temperature distribution in 127 µm thickness of Polyimide (Kapton) film was simulated using the pulsed laser irradiation with the 1064 nm wavelength and the Gaussian beam profile. The finite element method was used to solve the transient heat conduction equation (2) [43]. The 3D model was built for the Gaussian laser beam heat source (3) [44], assuming that 80 percent of laser energy was absorbed by linear absorption. The heat source was applied for 200 pulses scanned in the line with an X_1_ shift of 1 µm, as shown in Figure 10.
(2)$$\rho C\frac{\partial T}{\partial t}=\nabla \left(\kappa \nabla T\right)+Q$$

ρ is the material density (1.42 g/cm^2^ for Polyimide), κ is the thermal conductivity (0.12 W/m*K·for Polyimide), Q is the heat source, T is the temperature, C is the thermal heat capacity, t is time.
(3)$$Q_{n}=\frac{P}{f\times t_{p}\times {\omega_{0}}^{2}\times \mathrm{pi}}\times \alpha \times \left(1-R\right)\times \exp{\left(-\frac{x-x_{n}}{\omega_{0}}\right)}^{2}\times \exp{\left(-\frac{y-y_{n}}{\omega_{0}}\right)}^{2}\times \exp\left(-\alpha \times z\right)\times \mathrm{rec}1\left(t/t_{p}\left[\frac{1}{s}\right]\right).$$

Here, α is the absorption coefficient of polyimide, P is the laser power, x_n_ and y_n_ are the shifts of the coordinate depending on the subsequent position of the pulse, ω_0_ is the laser beam radius (at energy level 1/e^2^, it was 216 µm), z is the coordinate in the laser beam propagation direction, t_p_ is the pulse duration (10 ps), rec1(t) is the rectangular function to turn off the heat source, n refers to the pulse number. The modeling was performed when the applied irradiation dose to the sample surface was 7.34 J/cm^2^. Surface treatment with this irradiation dose led to the formation of LIG on the sample surface (see Figure 8). A schematic illustration of pulse laser beam translation is given in Figure 10. 

The time interval between pulses was 10 µs. The temperature distribution right after the 200th pulse (where the beam was translated by L_1_ equal to ω_0_) is presented in Figure 11.

From Figure 11, it is apparent that the surface of PI is kept heated above 1000 K for a time enough to irradiate the surface with 200 laser pulses (~2 ms), and to move the laser beam by a distance equal to ω_0_. The time interval of 2 ms of maintaining a temperature above 1000 K is enough to transform the surface layer of polyimide to graphene.

### 3.5. Electroplating

The Cu deposition on laser-treated Kapton PI films was performed using galvanic plating in an acidic copper bath (Figure 2). Plating was performed on a 10 × 20 mm rectangle laser-treated area. The applied current density was the main parameter for the electroplating. The optimized plating current and electrolysis conditions on different types of surface treatment were determined from the voltammogram of Cu deposition on laser-treated polyimide (Figure 12). The current density that could be applied for plating depended on the sheet resistance of the treated surface. An increase in the treated-surface sheet resistance resulted in a decrease in cathode current that could be applied. When sheet resistance increases, the Cu electrodeposition curve shifts to the negative side (Figure 12). The negative shift increases the overpotential of the Cu electrodeposition at a constant current density. For example, during the plating on LIG structure (8 Ω/square, curve 2, Figure 12), the current density of 10 mA/cm^2^ was at a low voltage of −0.17 V, in contrast, plating on the surface with a sheet resistance of 100 Ω/square (curve 4, Figure 12) was achieved when voltage was −0.34 V. During the plating on higher sheet resistances than 100 Ω/square (curves 5–7, Figure 12), a significant decrease in plating current was observed. Selective copper deposition on the laser-treated surface was possible when the treated surface sheet resistance was 10 kΩ/square or lower. The best plating results were achieved when plating on LIG (curve 2, Figure 12). It was most similar to copper-on-copper plating (curve 1, Figure 12). 

Three types of surfaces were formed on the PI sample after laser treatment: soot when fabricating with a high-intensity beam at the focal plane and up to 3 mm below the focal plane. Surface sheet resistance varied at these irradiation doses from 20 MΩ/square to 30 Ω/square (curves 3–7, Figure 12). Clusters of gray-colored structures formed after treatment in the range of 3.25–4.75 mm below the focal plane. Sheet resistance on the surface of these structures was 70–80 Ω/square. LIG structures formed when the sample was placed from 5 mm to 12 mm below the focal plane, with the sheet resistance of 8 Ω/square (curve 2, Figure 12). Selective copper deposition on the laser-treated surface was possible when the treated surface sheet resistance was 10 kΩ/square or lower. However, plating the whole treated area with a uniform layer of copper was possible only when the sheet resistance of the sample surface was lower than 100 Ω/square. Therefore, the thickness of a copper layer of 10–20 μm was required to obtain full coverage of the laser-treated area on the samples treated at the focal plane where soot formed (sheet resistance of formed soot was 30–100 Ω/square). Copper plating on LIG (sheet resistance of 8 Ω/square) formed a thin and homogeneous copper layer. Therefore, a minimal thickness of 5–7 μm only on LIG was needed to fully cover this type of laser-treated area (Figure 13). Plating on clusters of gray-colored structures was not performed due to improper fabrication quality.

## 4. Conclusions

Laser irradiation–polyimide surface interaction and galvanic plating on laser-treated areas were investigated in this study. Treatment with picosecond laser pulses was found to be more suitable for polyimide surface modification to electro-conductive material than nanosecond laser. Lower sheet resistance could be achieved after fabrication with picosecond pulses, and no sample deformations, due to the heating effects observed in this regime. 

It was shown that laser process parameters significantly impact sheet resistance. Treatment with a 1064 nm focused, high-intensity beam led to soot formation on the sample surface, of which the sheet resistance was strongly dependent on the irradiation dose and could be adjusted by changing laser power. Increasing irradiation dose in this regime led to the formation of a thicker soot layer, thus resulting in lower surface sheet resistance. Fabricating PI samples with a low-intensity defocused laser beam led to the formation of a smooth, gray-colored graphite-like structure, with an excellent sheet resistance of 8 Ω/square. Raman spectra of this structure showed a high intensity sharp 2D peak, which suggested the presence of LIG. The formation of LIG depended not only on the applied irradiation dose, but the irradiation (heating) time was also crucial. Heat conduction modeling revealed that the condition needed for LIG formation was to keep the polyimide surface temperature above 1000 K for more than 2 ms. 

Copper plating on the laser-treated surface was possible when the surface sheet resistance was 10 kΩ/square or lower. The electroplating process was faster on Kapton PI samples, treated in the defocus regime when LIG structures with a low sheet resistance of 8 Ω/square formed. The lowest copper layer thickness required to electroplate entire the laser-treated area was 5–7 μm on this type of sample. In contrast, to obtain the full copper coverage of samples treated at the focal plane, a 10–20 μm thickness copper layer was required. 

## Figures and Tables

**Figure 1 polymers-14-00781-f001:**
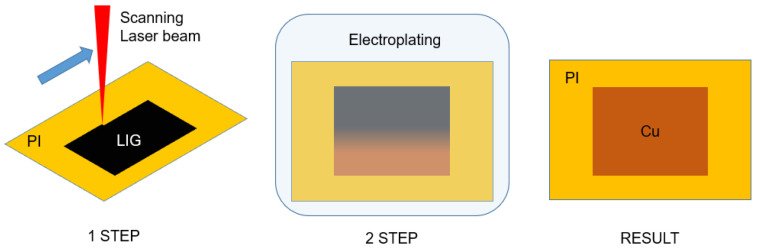
Schematic of two-step copper deposition process on polyimide (PI).

**Figure 2 polymers-14-00781-f002:**
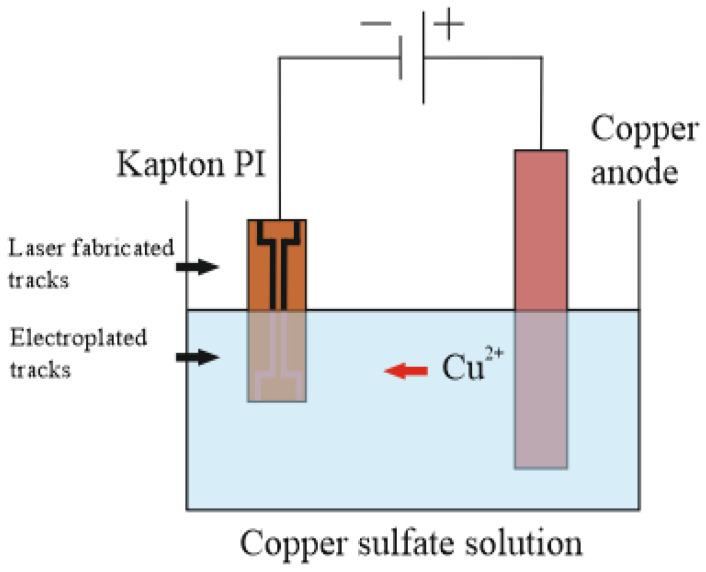
Electroplating scheme.

**Figure 3 polymers-14-00781-f003:**
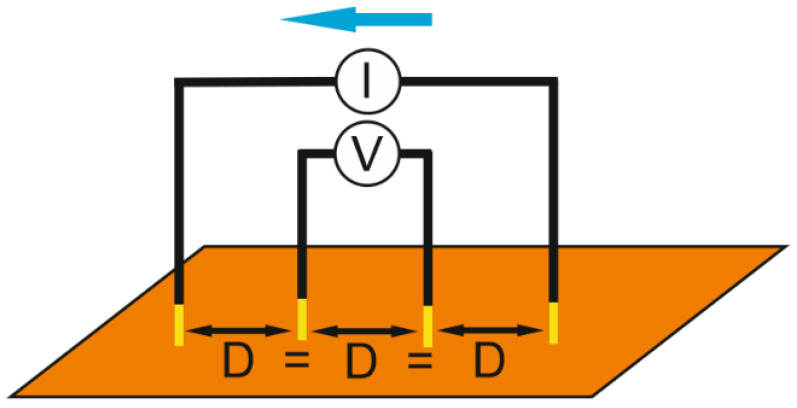
Sheet resistance measurement setup.

**Figure 4 polymers-14-00781-f004:**
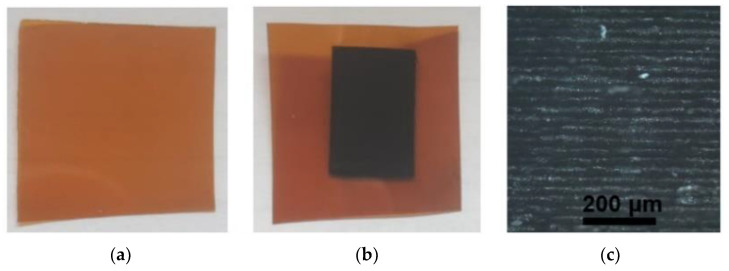
Polymer Kapton PI film: (**a**) before laser treatment, (**b**) after laser treatment and (**c**) magnified (×10) view of the laser-treated area.

**Figure 5 polymers-14-00781-f005:**
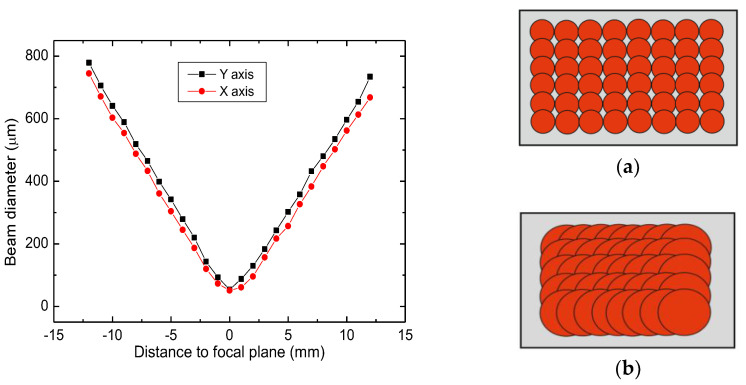
Beam diameter out of focus. At the focal plane, almost non-overlapping spots were obtained (**a**). Increasing defocusing, overlap increased due to significantly larger beam diameter (**b**).

**Figure 6 polymers-14-00781-f006:**
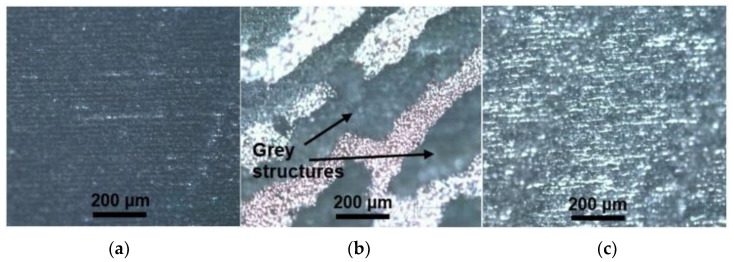
Microscope images of the sample surface: (**a**) treated at the focal plane. Soot appeared on the surface of the sample; (**b**) treated 4 mm below the focal plane, where clusters of gray-colored structures formed; (**c**) treated 7 mm below the focal plane. LIG structures, with smooth surface and low sheet resistance, were formed in this regime.

**Figure 7 polymers-14-00781-f007:**
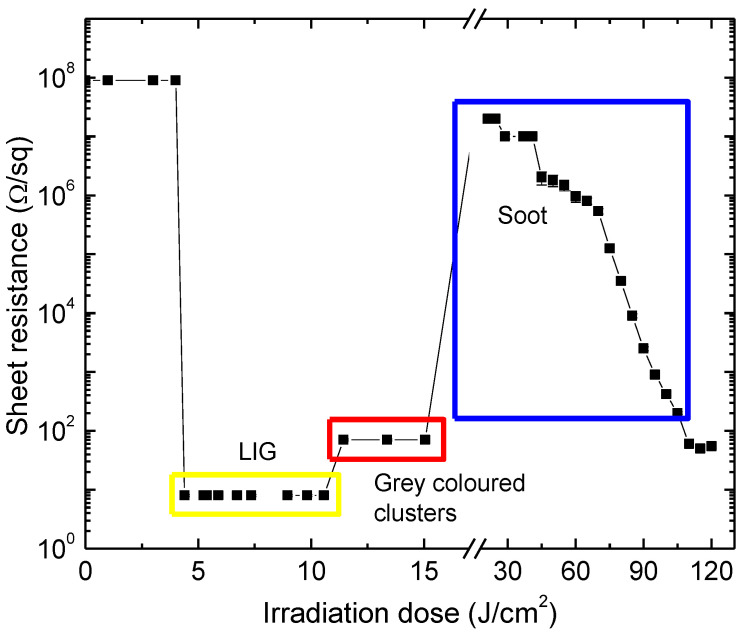
The sheet resistance of laser-treated polyimide surface versus irradiation dose. Irradiation dose was varied only by changing beam diameter on the sample surface. Three different polyimide surface modification regimes were distinguished. Depending on the applied irradiation, dose, soot, graphene (as we suggest LIG) or transitional state were found on the PI surface.

**Figure 8 polymers-14-00781-f008:**
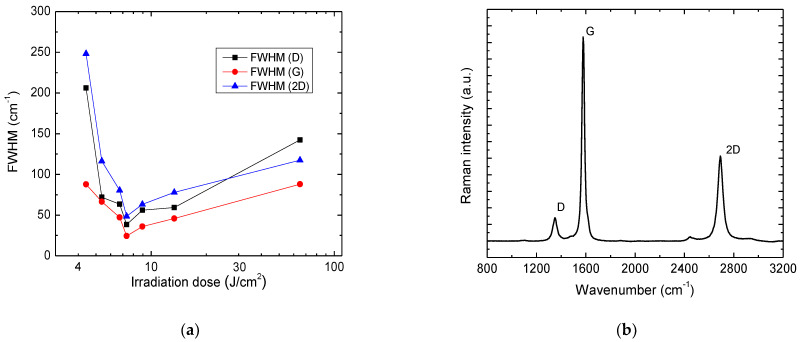
Dependence of FWHM of D, G and 2D bands on the irradiation dose (**a**); Raman spectra of LIG in PI achieved using the 7.34 J/cm^2^ irradiation dose (**b**).

**Figure 9 polymers-14-00781-f009:**
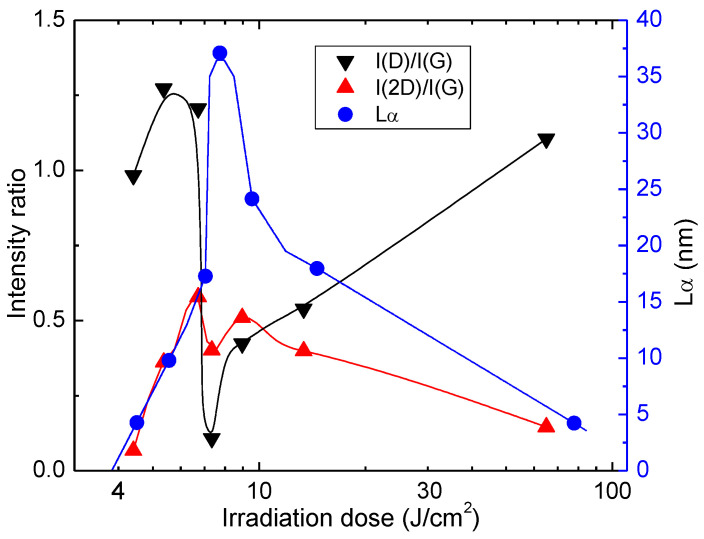
Dependence of I(D)/I(G) and I(2D)/I(G) intensity ratios and crystallite size $L_{\alpha }\;$ on laser irradiation dose at 1064 nm.

**Figure 10 polymers-14-00781-f010:**
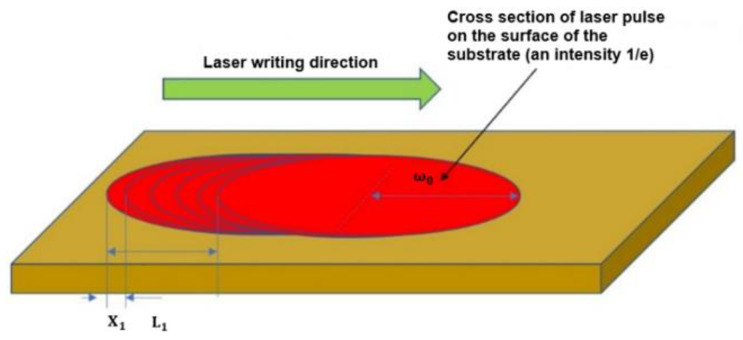
The schematic illustration of laser beam translation, used for thermal modelling. Here, X_1_ is the subsequent pulse translation distance, L_1_ is a distance of translation equal to ω_0_.

**Figure 11 polymers-14-00781-f011:**
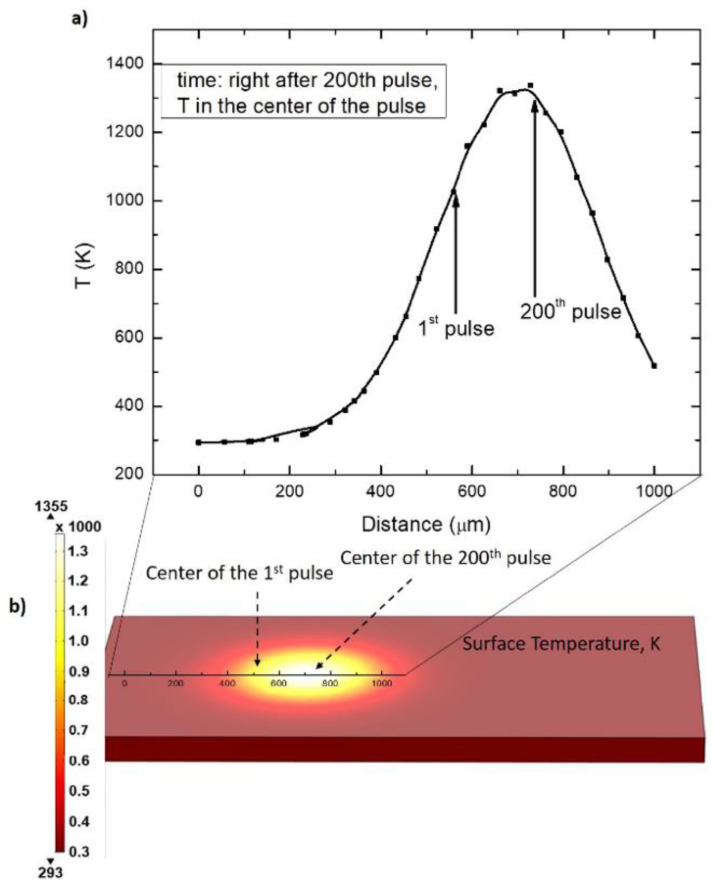
Temperature distribution right after the 200th pulse (where the beam was translated for a distance equal to ω_0_). (**a**) the temperature distribution in the center of the beam starting at the position of the 1st pulse to the 200th pulse at the time moment—right after the 200th pulse. The (**b**) represents surface temperature distribution at the time moment—right after the 200th pulse.

**Figure 12 polymers-14-00781-f012:**
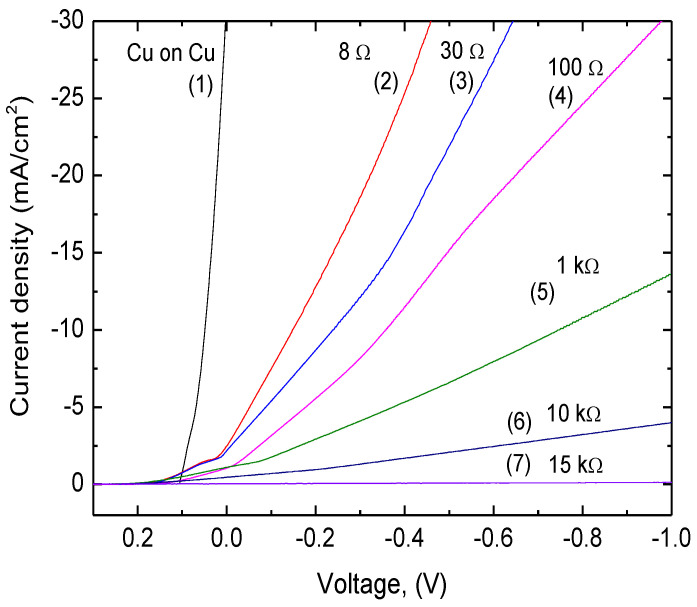
The voltammograms of Cu deposition on laser-treated surfaces with different sheet resistances. The voltammogram of Cu on Cu was also performed as a reference. The potential sweep rate was 2 mV/s and the potentials were given versus Ag/AgCl/KCl(sat) electrode. The measurements were carried out at room temperature using the potentiostat/galvanostat Reference 600 (Gamry Instruments, Warminster, PA, USA).

**Figure 13 polymers-14-00781-f013:**
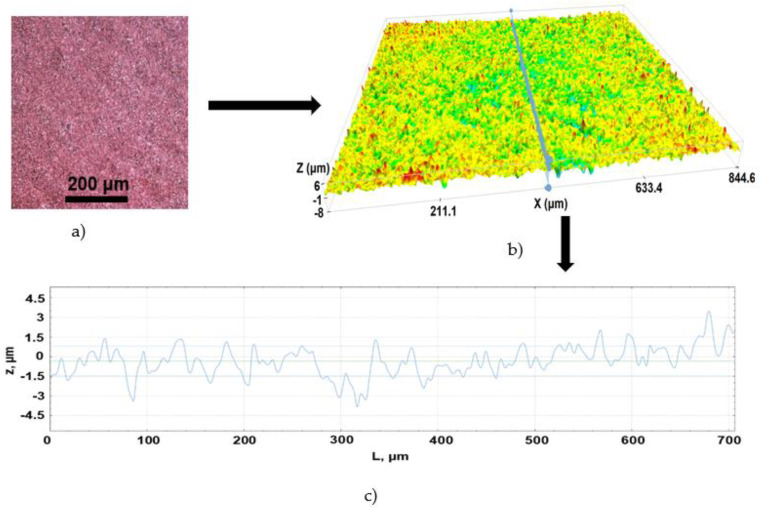
(**a**) Homogeneous copper layer plated on the LIG surface (magnified x10). (**b**) 3D optical profilometer image of copper-plated sample surface topography and the line shows 2D surface topography along the middle of the image (**c**).

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
