# Peer review of "Laser-Assisted Selective Fabrication of Copper Traces on Polymers by Electroplating"

_polymers, 2022, doi:10.3390/polym14040781_

Round 1

Reviewer 1 Report

This manuscript describes the Laser-assisted selective fabrication of copper traces on polymers by electroplating. The work should be in the interest of the reader of the journal. There are, however, a number of points to be considered by the authors prior to the possible publication of this paper. In short, the manuscript needs thorough revision in terms of language and clarity. The queries raised above also need to be addressed. Thereafter only the manuscript can be reviewed meaningfully. Hence the recommendation is to thoroughly revise the manuscript.

  1. The author should be revised the title and make it more attractive.
  2. Conclusions and abstract sections are not clear. Write very concise and specific.
  3. Lot of typographical errors and grammatical mistakes were noticed in the manuscript. Need to complete for check for the entire manuscript.
  4. Introduction section shows to discontinue, so the author should be revised and make it more attractive.
  5. Author should use the same format of all figures (figure 2).
  6. The main advantages and disadvantages of the present study must be added.

Author Response

We are grateful to the reviewers for their valuable work in careful reading of our manuscript and providing comments which, we hope, made our manuscript better understandable by readers. Please see the attachment with response to the reviewer's comments.

Reviewer 2 Report

The authors proposed a novel method for enhancing the electroplating of polyimide films by pre-treatment using laser radiation. The preferential electroplating of copper appears on the places where  a graphene layer has been created by laser irradiation. The paper is well written and the flow of the argument is smooth. I suggest to expand the discussion of the advantages and disadvantages of the presented method in comparison with alternative laser-based technologies. There is also a repeating error on lines 87, 146, 188, 263 etc. --> Please correct.   

Author Response

We are grateful to the reviewers for their valuable work in careful reading of our manuscript and providing comments which, we hope, made our manuscript better understandable by readers. Please see the attachment with the response to the reviewer's comments.
